



# Towards family-friendly conferences in the geosciences: results from a first survey

Elena Päffgen[1], Lisa Schielicke[1,2], and Leonie Esters[1]

[1]Section Meteorology, Institute of Geosciences, University of Bonn, Auf dem Hügel 20, 53121 Bonn, Germany
[2]Department of Physics and Astronomy, The University of Western Ontario, London, Ontario, Canada

**Correspondence:** Elena Päffgen (s6elpaef@uni-bonn.de)

**Abstract.** In the geoscientific field, building an academic career requires a high level of dedication to research, frequent publishing, and maintaining visibility within the academic community. Conferences are the central platforms for networking and knowledge exchange, making them keystones for a scientific career. However, the early stages of an academic career often coincide with family planning considerations. Many young researchers face the challenge of balancing professional responsibilities with family obligations. A balancing act that becomes particularly difficult during non-routine work events such as conferences. Such occasions tend to pose significant organizational challenges for families. As mothers still perform most caregiving, they are disproportionately affected. This conflict is an important factor inhibiting the rise of women through the academic career ladder. Hence, increasing the family friendliness of conferences holds a chance of diversifying the community, allowing more women to participate, while reducing the tension of the academic work-family conflict.

In this work, we present the results from an online survey among researchers from the geoscientific field. The required support of parents at conferences is analyzed and the acceptance of researchers without children for the implementation of new support offers is examined. Our key findings are that (1) parents wish for more support at conferences, (2) the majority of non-parents welcome family-friendly measures and (3) conference organizers can accomplish a lot in this respect with small adjustments. The responses we received from parents paint a qualitative picture of measures that can be taken to create a family-friendly conference experience, ranging from an increase of transparency and awareness for the challenges of parents to tangible offers like childcare and financial funding. This feedback invites conference organizers to join the conversation about family support and shows that new measures have great support in the geoscientific community. Our findings are distilled into guidelines for conference organizers, allowing them to better address the work-family conflict and transform their conferences towards more equity, diversity and inclusion.

## 1 Introduction

Participation in scientific workshops and conferences is essential for academic growth and offers opportunities for knowledge exchange, skill development, and professional networking. For early-career researchers, these events play a vital role in establishing their presence within the scientific community, refining presentation techniques, and initiating collaborations. Networking at conferences facilitates connections with potential employers, initiates future collaborations, and supports securing



research funding. According to Hauss (2020), young researchers emphasize that conferences strengthen professional networks
and increase publication opportunities, while they also foster a sense of visibility and identity as a scientist.

Despite progress in gender equity, women remain underrepresented in science, technology, engineering, and mathematics
(STEM), particularly in postdoctoral positions and beyond (Vieira do Nascimento et al., 2021). Balancing academic careers
with family responsibilities is especially challenging for young parents, with women often bearing a disproportionate share of
caregiving duties (Cattaneo et al., 2019). Studies have shown that work-life balance constraints can limit career advancement
for women in STEM. For instance, Thoman et al. (2022), who interviewed women in STEM fields from the southwestern
United States, found that 80% face a choice between delaying or forgoing parenthood or reducing their workload, e.g., by
opting for less ambitious career paths. This is consistent with findings from Ceci and Williams (2011) and Baker (2010), which
underscore the higher likelihood of women to seek part-time or flexible roles to achieve work-family balance. In this work, the
terms caregiver and parent are used interchangeably to refer to a person that is responsible for the well-being and upbringing
of a child.

The timing of family planning often coincides with critical career stages. In STEM, the average age for earning a PhD is
approximately 30 years (Statistisches Bundesamt, 2021a; National Center for Science and Engineering Statistics (NCSES)),
2021), while the average age of first-time parents in Europe is 29.7 years for women and 33.2 years for men (European
Commission, Eurostat, 2022; Statistisches Bundesamt, 2021b). In the U.S., parents tend to be slightly younger, with women
giving birth to their first child at 27.3 years (National Center for Health Statistics, 2021). These overlapping timelines create
unique challenges for early-career researchers who are parents, particularly when it comes to conference participation. Without
adequate support, caregivers are often excluded from these crucial networking and professional development opportunities.
Callaghan (2016) observed that women with dependent children are significantly less likely to present at conferences.

Although the work family conflict interferes with scientists' ability to start a family, their desire to have children remains
strong. In Germany less than one-third of scientists have children (Metz-Göckel, 2014), but almost 80% express a desire to
have them (Gottburgsen et al., 2022). A very similar picture can be seen in the US, where scientists have been found to have
fewer children than they want when pursuing a scientific career (Ecklund and Lincoln, 2011). However, the discussions about
family responsibilities remain scarce within the scientific community. This discrepancy underscores the need for more open
conversations and support for parents in academia.

Attending conferences as a parent or caregiver includes navigating logistical, financial, and emotional challenges. Lack of
family-friendly policies can limit opportunities for caregivers to participate, which ultimately leads to inequities in academia
(Angela L. Bos and Schneider, 2019). Addressing these barriers is critical to advancing equity, diversity, and inclusion (EDI)
in STEM fields. Family-friendly policies have been shown to enhance inclusiveness and retention, as demonstrated by Singh
et al. (2018), who found that organizational efforts to support a work-life balance significantly increase women's attachment to
their field and their likelihood of persisting in their careers.

There is a clear need for family-friendly conferences, since creating such inclusive environments is essential to fostering a
more diverse and equitable scientific community (Swann, 2019). Gender-diverse teams are known to produce better science
(Nielsen et al., 2017), and family-friendly policies can help ensure broader participation from parents and caregivers. While



the geosciences community has recognized the importance of such policies (e.g. The Geological Society of America, 2023),
data on their prevalence and effectiveness remain limited. This highlights the need for systematic research to identify effective
strategies that support caregivers at scientific events.

This study presents findings from an online survey conducted in 2023 as part of a project on family-friendly conferences
in the geosciences. They were first presented at the EGU General Assembly 2024 (Päffgen et al., 2024). The survey aimed to
explore the challenges faced by parents and caregivers at academic events and to gather insights on potential support systems.
Drawing inspiration from Pohl et al. (2022), who developed a checklist for family-friendly conference culture, this survey
sought input from scientists with and without children to identify practical measures for conference organizers.

The study addresses the following questions:

– How can parents be supported at scientific conferences?

– What do scientists without children think about family-friendly offers at conferences?

– What actions can conference organizers take to ensure that their event is family-friendly?

Based on our findings, we propose guidelines for organizing family-friendly conferences, including a practical checklist that
is divided for times before, during, and after the event.

This paper is organized as follows: Section 2 describes the structure and design of the survey. The results of our survey are
presented in Section 3. Section 4 provides an analysis of the results and introduces guidelines for enhancing family-friendliness
at conferences. A detailed checklist for organizers is provided in Appendix A.

## 2   Methods

The survey focused on potential support systems for increasing the family friendliness of scientific conferences. To that end,
ideas and opinions from geoscientists with and without children were collected. Our survey design was guided by Pohl et al.
(2022), who interviewed participants at a meeting of psychologists about family status, childcare, and support needs. Their
survey lead to a checklist for improving family-friendly culture, communication, and logistics at future meetings. Our study
expands on the work of Pohl et al. by considering a broader sample that includes international researchers attending a variety of
conferences in the geosciences. Consequently, this allows us to create a more comprehensive guide for conference organizers.

The survey was created with the online tool Empirio (2024) and was online and open for participation from 30 June until
31 October 2023. During that time, it was advertised on several social media platforms such as X and LinkedIn as well as
in newsletters of geoscientific institutes and at several international conferences. The survey was distributed via networks of
female and early career scientists. The survey was available in English and German. Of the 245 scientists that participated, 101
participants conducted the English and 144 the German version of the survey. The survey contained 22 questions in total: 18
addressed caregivers and 14 non-caregivers. A complete overview of the questions is given in Tab. B2 in the Appendix B. The
first three questions focused on general information on conference participation. Depending on whether a participant stated to
have children or not, they were directed to a different set of questions. While scientists with children were asked about their



needs and experiences at conferences, scientists without children were asked about their opinion on increasing support for families at conferences (see Fig. 1). In the final part of the survey, both groups were asked demographic questions about their age, gender and status group (see Tab. B1). Most of the questions were optional. On average it took the participants six minutes

to complete the survey.

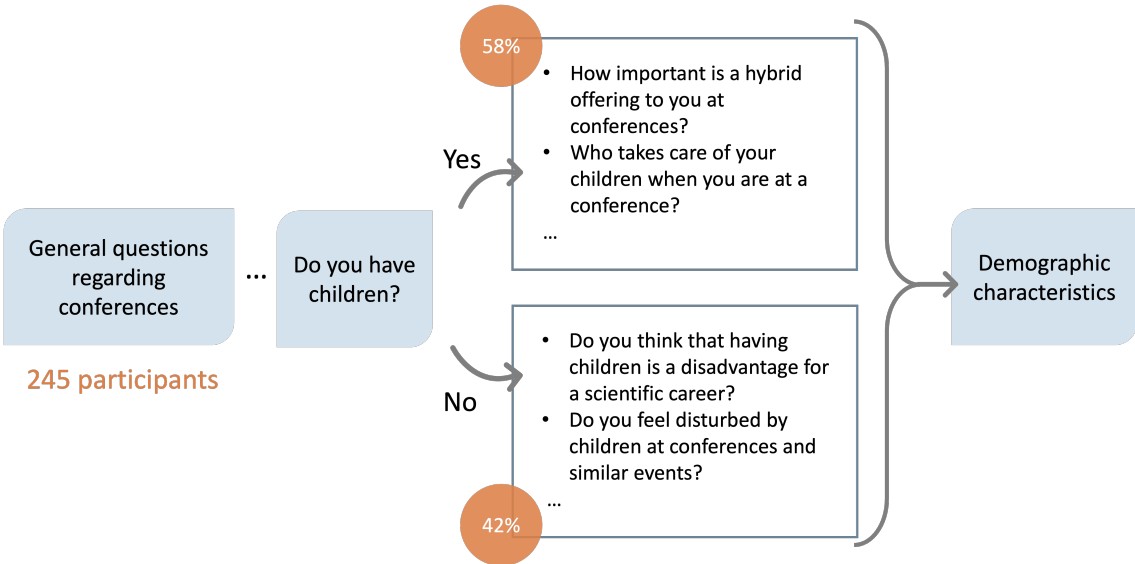

**Figure 1.** Structure of the online survey.

In the survey, different question formats were applied. The majority of questions were in the single or multiple choice format, giving the participants the opportunity to choose from an array of options. The answers to these questions were used for a quantitative analysis. Moreover, to give caregivers the opportunity to include their own ideas, feelings, and experience of family-friendly conferences, a free text question "What would you wish for to increase a family-friendly environment on

conferences or workshops?" was included in the survey. Answers to this open question were analyzed using a qualitative content analysis by inductive category formation following Mayring and Fenzl (2019). We used the web based tool QCAmap (Association for Supporting Qualitative Research ASQ, 2020) and categorized the answers with the analysis question "What do parents wish for to increase the family-friendly environment on conferences?" by going through them line by line. The categories were formed inductively based on the answers given. The software QCAmap provided an overview of how many

answers were sorted into which category (AppendixC). Based on the categories, the importance of the mentioned topics were evaluated. To validate our results, we performed an intercode agreement check (Mayring and Fenzl, 2019). A non-involved student assistant applied the same analysis to gain an independent perspective on the categories. Finally, the results were compared to hers (Fig. C1).



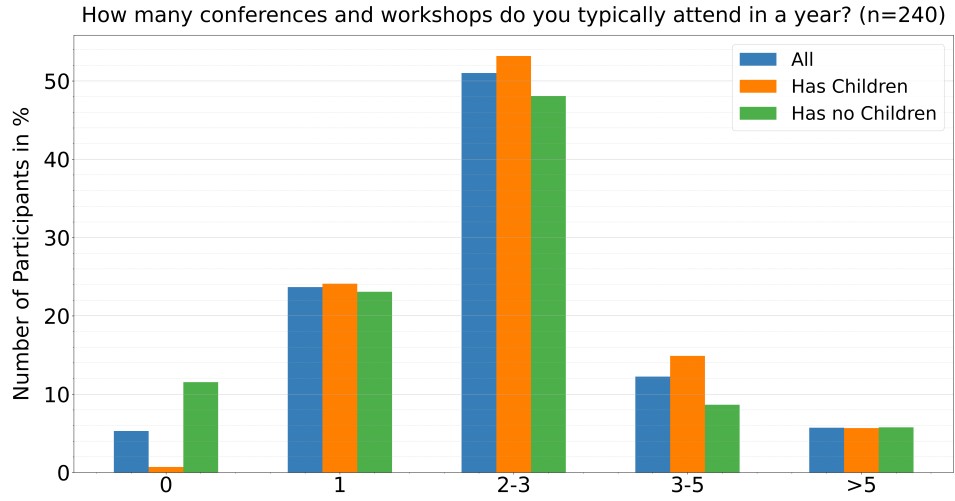

**Figure 2.** Answers on the question 'How many conferences and workshops do you typically attend in a year?' Only one answer was possible. This question was posed to all participants (245) and skipped by five participants. 240 participants answered this question.

## 3 Results

The analysis of our survey followed two main questions: (1) How can caregivers be supported at scientific conferences? (2) What do scientists without children think about family-friendly offers at conferences? Based on our survey, we aimed to answer these questions qualitatively. In addition, based on the survey, guidelines for conference organizers are proposed.

### 3.1 Demography

The survey was completed by 245 scientists with 56% identifying themselves as female. Most of the participants (73%) were between 25 and 44 years old. The participants came from various academic backgrounds and attended a diverse range of conferences, 69% visiting two or more conferences per year (Fig. 2). A slight majority of 58% of all participants were caregivers. These caregivers had one or more children in a range of ages that we grouped into children younger than 3 years (37% of all caregivers), 3-6 years (46% of all caregivers), 7-10 years (35% of all caregivers) and older children (33.4% of all caregivers). Therefore, each age group represented at least one third of all caregivers.

One participant identified as non-binary and one selected the option "Other". In addition, nine participants skipped the gender question. When analyzing gender dynamics, we did not include these groups because their numbers were too small.





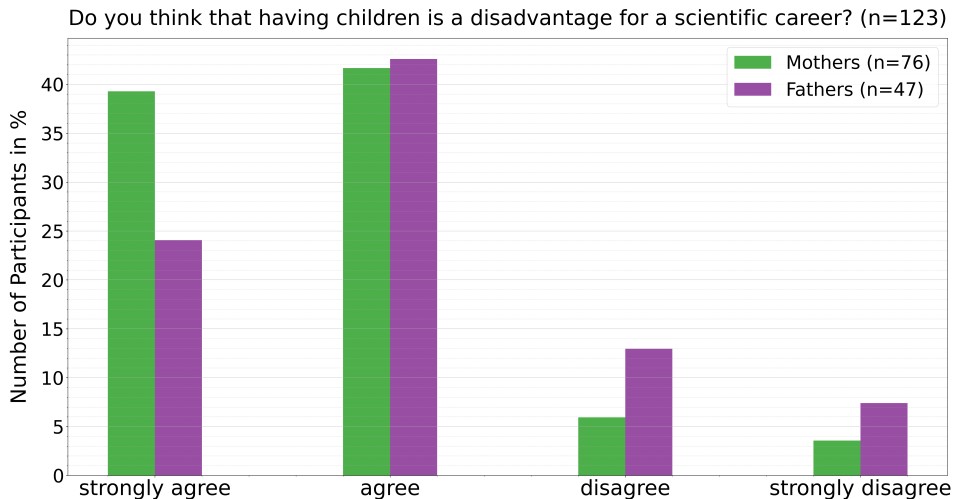

**Figure 3.** Answers to the question "Do you think that having children is a disadvantage for a scientific career?". 126 parents answered this question (skipped by 15). The answers are sorted into the indicated gender. The shown percentages correspond to the respective group.

### 3.2 How can parents be supported at conferences?

The need for support for families at conferences was particularly evident in the survey: 78.3% of the mothers and 60.0% of the fathers stated that they could not attend a conference due to family obligations in the past. Furthermore, approximately 77% of all participants agreed or strongly agreed with the statement that having children was a disadvantage for a scientific career. Here, mothers feel particularly often that children are a disadvantage to their career compared to colleagues without children (Fig. 3).

Based on the parent's answers to the question "What would you wish for to increase a family-friendly environment on conferences or workshops?" using inductive category formation, we deduced the following four categories for family support:

(1) Family-friendly infrastructure at conferences including (a) Possibilities for exchange and networking for parents, (b) Events, where Children are explicitly Welcomed, (c) Family-friendly event schedule, (d) Childcare and rooms explicitly designed for families, (e) Hybrid events

(2) Financial support

(3) Awareness

(4) Clear communication and transparency



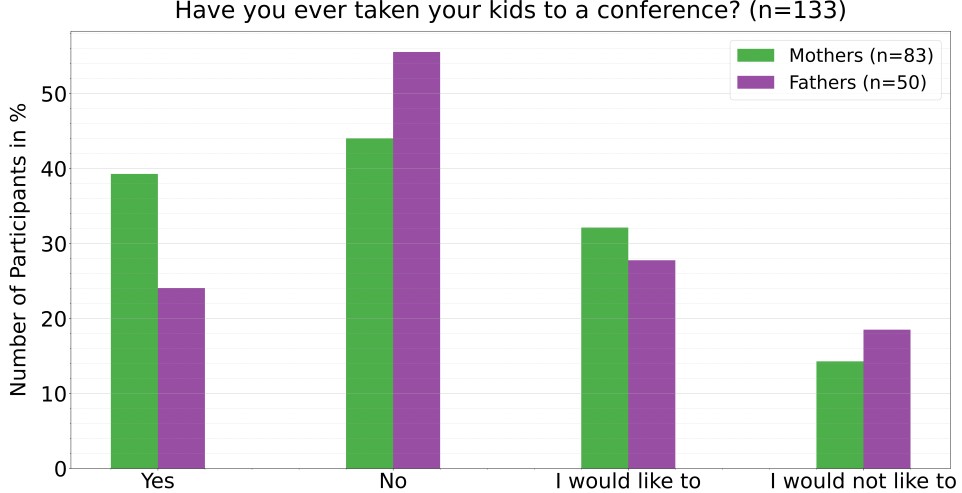

**Figure 4.** Have you ever taken your kids to a conference? Multiple answers possible. Only participants with children answered this question (134 participants in total). Skipped by 7. The answers are sorted into the indicated gender (only answers from mothers and fathers included here). The shown percentages correspond to the respective group.

**(1) Family-friendly infrastructure at conferences**

The category of family-friendly infrastructures focused on the support offered during the conference and at the conference site. As shown in Fig. 4, 43% of the mothers and 24% of the fathers stated that they have already taken their children to conferences. In this respect, it is striking that the number of mothers is almost 80% higher. At the same time, 32% of all mothers and 28%
of all fathers would like to bring their children to future conferences. Figure 5 summarizes the relative importance of family-friendly infrastructure offerings to the parents interviewed. Some parents prefer to attend conferences without children. Those caregivers stated "I prefer going without my child to conferences for networking." and "Talks are important, but networking is more important.".

To allow caregivers to bring their children, inclusive scheduling of talks and events is the way to go, but it is equally important
to communicate to parents explicitly when they are welcome to bring their children to an event. Furthermore, it is important to recognize that parents, as a group, can provide an invaluable support structure for each other. Therefore, organizers may find it wise to use these existing resources by creating forums that specifically target parents. Networking is a high priority for caregivers, of whom 39% named it their main concern (Fig. 5).

Apart from integrating children into their parents' conference activities, it is important to provide them with their own spaces,
e.g., a room where they can do their homework or even dedicated events, where the research is presented to children in an age-appropriate manner. While older children are well served with the aforementioned offers, young children require childcare. Childcare service was rated as helpful by 71% of the caregivers interviewed (Fig. 5). Especially at international conferences,



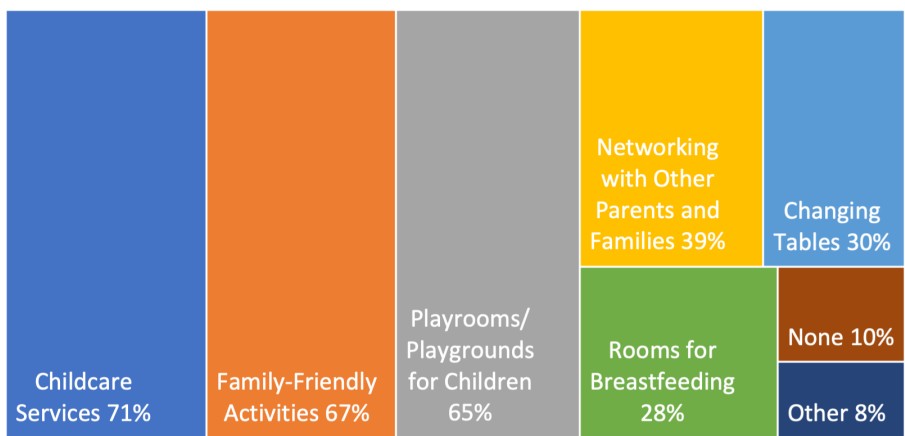

**Figure 5.** Hierarchy plot of the answers to the question "Which of these offerings might help you at conferences and similar events?". Multiple answers were possible. Only participants with children answered this question (139 participants in total). Skipped by 2.

children are going to speak different languages. To make all children feel welcome, it would be ideal to have babysitters who are fluent in several languages. In this context, it is crucial to provide an environment that feels safe for both the parent and their
child. Another important consideration is that conference schedules and the specific needs of children are individual, which should be reflected in a flexible daycare that allows children to drop off and pick up at different times. Additionally, rooms for families that serve other purposes, for example breastfeeding rooms are voted helpful by 28% of parents.

When asked about the possibility of online participation, 76% of the parents rated hybrid events as important or very important. While they acknowledged that this would mean losing out on some of the conference experience, an online participation
seemed, for many of the interviewed caregivers, to be the only option to attend at all.

**(2) Financial support**

Even though it is convenient to attend an event online, sometimes it is crucial to attend a conference in person. To enable participation for parents, a financial infrastructure along with the digital one is needed. Of the 66 responses from parents to the question "What would you wish for to increase a family-friendly environment on conferences or workshops?", 29 included
requests for financial assistance from institutions and conference organizers. Some participants stated that financial support for families, when available, is often too low and comes with extensive documentation. Other parents complained that bringing their partner to take care of the child to the conference means that the partner has to take leave for the travel and usually must pay the additional travel and accommodation costs. Conference fees for children or caretakers were said to add to these costs. Also, childcare offered at the conference site was claimed to be expensive at times. Scientists who choose to leave their child
at home supervised by the partner or a caretaker said they must either bear the costs for extra hours of daycare, or the partner




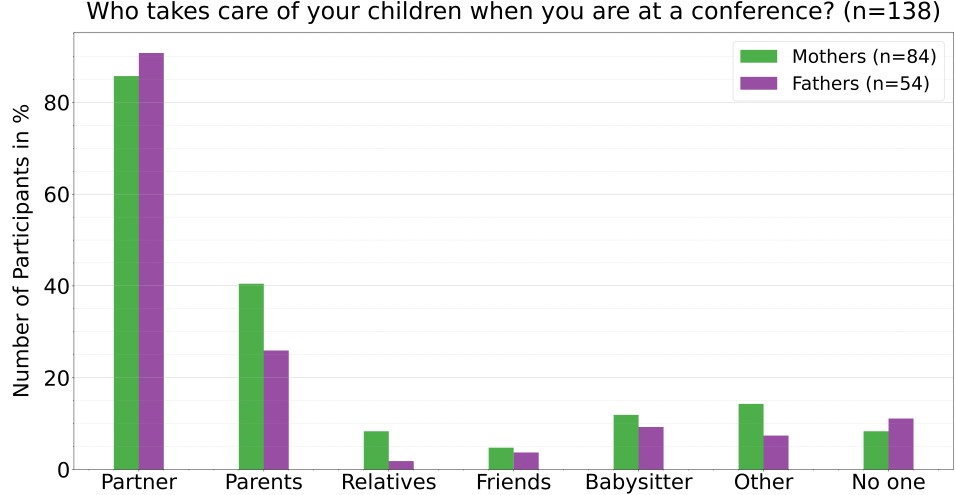

**Figure 6.** Answers to the question 'Who takes care of your children when you are at a conference?' Only participants with children answered this question (139 participants in total). Skipped by two. Only the answers of parents indicating their gender as "female" or "male" are shown.

would have to take leave. Here, in particular family members are taking up support tasks. With 92% of all fathers and 87% of all mothers, parents rely heavily on their partner when attending a conference (Fig. 6). 48.8% of women with children state that they sometimes resort to their parents or other relatives and 13 % resort to a babysitter for care taking. Fewer fathers use these options with 27.8% and 8%, respectively (Fig. 6).

**(3) Awareness**

Parents often feel isolated and overwhelmed by their role as caretakers within the academic world. One participant expressed this sentiment with the comment, "Make it normal to have kids," reflecting the loneliness associated with balancing parental responsibilities.

Scientists in our survey felt that they are missing out when unable to participate in conferences as fully as their colleagues
without children. This sense of being left behind by the scientific community places significant pressure on parents, as one respondent stated: "I can already tell that I'm being left behind by the international community."

**(4) Clear communication and transparency**

Life situations and needs of families are diverse. Family-friendly conference organization therefore requires good and ongoing mutual communication, as well as individual and creative solutions. The lack of transparency on family support options in-
creases the mental load and organizational effort that parents must bear prior to attending conferences. No matter how well the





family support system might be, it is useless if the information is not effectively passed on to the families who need it. Our survey revealed that 67.4% of parents do not feel well-informed about support offerings for families. To address this, the following information is crucial: First, it is essential to clearly define when and where children are welcome during different conference events, helping families plan their participation without unexpected challenges. Second, releasing the conference program well

in advance gives families enough time to organize their trip, arrange work leave, and apply for funding if they plan to bring their children. Third, it is essential to offer comprehensive details about the childcare services available during the conference, including hours of operation, qualifications of caregivers, prices, and how to register. The more detailed the information is, the more secure parents feel to leave their children at the provided daycare options. A list of activities available for families in the vicinity of the conference venue can be very helpful, offering them options to explore and enjoy their time together. The

survey showed that for several parents visits to the playground for example have improved their conference experience: "At the EGU, we went to the water playground on the old Donau at lunchtime, which was great compensation for the children cominpg back to the conference and sitting through a few presentations." (translated from German to English). In addition, sharing a list of affordable hotels that focus on families makes it easier to find a suitable accommodation. On-site, detailed information about the family-friendly infrastructure should be provided, including facilities like nursing rooms, play areas, and

restrooms equipped for young children. Plus, sessions that cover the family-work conflict or similar topics can get their own tag so that they are easier to find in the conference program. Including the option of adding a sticker to conference attendees name batch that indicates that they are a parent, can help parents at the conference to find each other and be a conversation starter. Beyond logistics, it is vital to explicitly communicate that families are welcome at the event. It raises awareness of people with family responsibilities and makes parents feel seen. This can already be achieved by a prominent statement on the conference

website or a mail with information for parents distributed among all attendees. Encouraging communication among attending parents before the conference, allowing them to connect and network based on shared experiences, can also help distribute the information about the support in place. Additionally, clarify how many tickets are needed if children or partners are brought along, ensuring families can plan their budget accordingly. Finally, it is important to communicate that special arrangements can be made for families with unique needs, offering flexibility and personalized support. This can be achieved, for example,

by setting up a dedicated e-mail address or telephone number. Being transparent also includes communicating, which support options are not available at the conference, so that parents have no reason to worry.

### 3.3 What do scientists without children think?

Including the perspectives of scientists without children in this discussion is essential for two reasons: First, to secure their support as allies, and second, because they may eventually become parents themselves. A transformation towards a more family-

friendly environment at conferences will impact all attendees. Therefore, it is crucial to implement the changes thoughtfully, ensuring that no tension arises between those with and without children. Participants without children were asked whether they felt disturbed by the presence of children at conferences. A majority of 71% indicated that they were not disturbed. However, 21% expressed that they feel disturbed. Therefore, it can be helpful to clearly designate where children are welcome, whether in sessions, lectures, workshops, or during free-time activities, and where they are not.



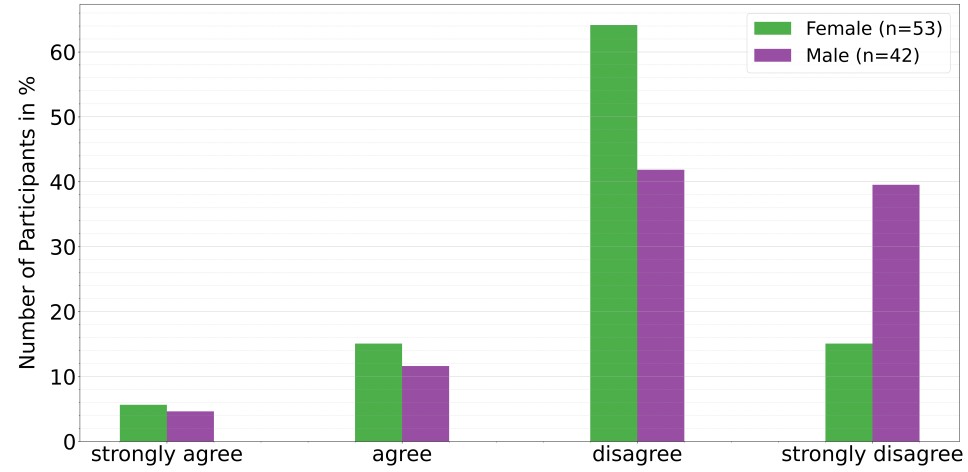

**Figure 7.** Answers to the questions 'Do you feel that conferences and similar events are designed to be family-friendly?' Only one answer is possible. Only participants without children answered this question (103 participants in total). Skipped by 1. Only the answers of female and male participants are included in this plot.

Most of the participants (79%) without children do not think that conferences are designed to be family friendly (Fig. 7). Male and female scientists disagree or strongly disagree with the statement that conferences are designed to be family-friendly. However, male participants leaned more towards strongly disagree. Overall, 84% of participants without children agreed or strongly agreed on the need to expand family-friendly offerings at conferences, indicating strong support from the broader community. This is important, as funding for such initiatives may come from the shared conference budget. Not only does

the majority favor increasing support for families, but 80% (79.06% of all male and 83.01% of all female scientists) are also willing to actively participate in these efforts (Fig.8). Such events could help researchers to gain a new perspective on their work and at the same time would be an opportunity to make the conference stay for children memorable.

**3.4 Composition of a guide for conference or workshop organizers**

Conferences still need to improve in the field of equity, diversity and inclusion. For example, multiple studies show that

engagement in discussions is dominated by senior men (Lefebvre and Bernhard, 2024; Jarvis et al., 2022; Gulland, 2022). Increasing family support at conferences can diversify the community, leading to more innovative results (Calisi and C. Lopes, 2018). Conferences play a critical role in promoting inclusion. However, addressing the needs of parents requires a high degree of flexibility, as their needs are highly individual. These vary with child age, family situation and the support provided by the researcher's home institution. It is important to note that not all needs can be met.

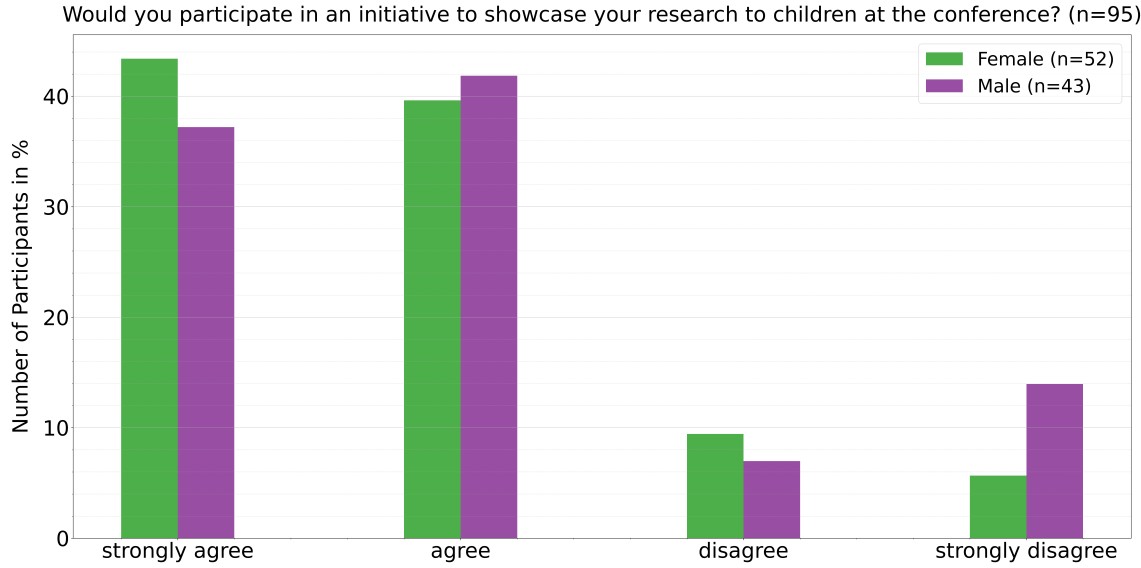

**Figure 8.** Answers to the questions 'Would you participate in an initiative to showcase your research to children at the conference?' Only one answer is possible. Only participants without children answered this question (102 participants in total). Skipped by 2. Only the answers of female and male participants are included in this plot.

There are conferences in the geoscientific field that already try to become more inclusive. However, since there are no unified guidelines, every conference takes its own approach, leaving parents in a maze of different programs. At the same time, the resources and options available to conference organizers vary significantly depending on the event format. International conferences face other challenges than national events. Conferences with varying locations need different approaches than conferences that occur at the same venue. To address these challenges effectively, it is crucial that each organizing committee appoint a dedicated commissioner responsible for family support. In general, conferences should make it a priority to talk to parents and subsequently incorporate their suggestions and feedback into the design of the event.

Based on our results, we propose a guide structured to progress from initiatives requiring minimal financial and organizational effort to those demanding greater resources and planning. The guide provides concrete, actionable ideas to ensure practical relevance for conference organizers. The recommendations are directly derived from our survey responses, addressing real-world challenges and needs of families. In Appendix A, the guide is presented as a check list organized by measures that can support families before, during and after the conference. Similar findings are presented by Saccarelli et al. (2020) in the field of radiology and by Calisi and C. Lopes (2018), wherein a working group of mothers proposes CARE guidelines to "tackle the childcare-conference conundrum".





## 4   Discussion

The survey results of our study show the urgency of family support in academia. The results show that balancing academic work and family responsibilities remains a major difficulty. Most of the scientists interviewed perceived having children as a disadvantage for a scientific career. Especially for women with children, the work-family conflict persists. This agrees with the finding of Santos and Cabral-Cardoso (2008) that mothers of dependent children are most affected by tensions between academic work and family life.

A range of very different responses showed that a family-friendly conference looks different for every parent. Within the results, we identified multiple facets of family support, namely "Family-friendly infrastructure", "Financial support", "Awareness" and "Clear communication and transparency". In agreement with Singh et al. (2018), the need for socio-emotional support as well as tangible support, such as childcare and funding, is evident in the answers of parents.

The possibility of exchange and networking plays an important role for parents as it is an integral part of the conference experience. There is an understanding among researchers that networking is important for a successful career in academia (Ansmann et al., 2014; Fisher and Trautner, 2022; Li et al., 2022; Hauss, 2020). Conferences are crucial to building and maintaining a network that can be beneficial for collaboration and publishing activities. Conferences usually offer dedicated networking events, such as coffee breaks, an ice breaker, or a reception and conference dinner. Since these events are often held in the evening and sometimes involve the consumption of alcohol, they often exclude scientists with children. Actively integrating

family life into conference events and creating dedicated spaces for families could mitigate the described disadvantages.

In addition, conference organizers should think about inclusive, hybrid or online options that replace in-person networking. Dedication to digitizing all of the conference events and increasing the virtual experience opens the door not only for families, but also for scientists with other care responsibilities, few financial resources or disabilities.

For many families having professional daycare options would be a significant help. Making them free of charge would be

a good way to accommodate parents financially. However, especially for scientists with young children and nursing mothers, giving them the possibility to bring their own babysitter or caregiver, i.e. a partner or relative, along, is even more important, since "[Children] need a reference person until a certain age." (Answer of a parent). Accompanying persons can be integrated by allowing them to enter free of charge. In parallel to daycare options, breast-feeding rooms and accessibility with strollers should be provided, because they affect those children who are the most dependent on their parents. In the spirit of inclusion,

changing rooms should be open to all genders. In general, rooms that are specifically dedicated to families would be helpful. These can serve a wide array of functions, from the mentioned breastfeeding rooms to a space where parents can watch their children and follow the conference via a life-stream. The survey showed that scientists without children are very much in favor of greater support for families, so that it would be realistic to consider all participants paying for the additional costs that arise with the childcare options. Increasing the flexibility in conference fee payments, so that it is possible to join an event for a

shorter period meets the needs of those parents that prefer attending the conference without their child. Conference organizers can explore the following policies to reduce costs for parents: (1) free childcare provided by the conference, (2) children and accompanying caregivers may attend the conference at no conference fee, (3) accommodation options for families are listed





on the conference website, and (4) travel subsidy for parents who wish to bring their children. These are the first steps to make several conference visits a year affordable for parents. Easy access to financial support is the key to a more family-friendly

environment at conferences.

The mental load that parents bear, particularly around non-routine work events like conferences, demands a significant amount of planning. Establishing networks for families can help parents feel less isolated. In this regard, creating a platform for sharing personal stories and experiences could foster greater sensitivity to these challenges within the scientific community. In addition, educational initiatives such as panels on balancing parenthood with a scientific career would raise awareness and

advance the conversation. Offering such sessions at conferences would likely lead to greater recognition of family responsibilities that would carry over into everyday work settings. This might give young women who start a degree in STEM a new perspective on their future - to see personal and professional goals as complementary (Thoman et al., 2022). This is supported by Santos and Cabral-Cardoso (2008) concluding that "work-family policies are fruitless, unless they are supported by a positive work-family culture". Furthermore, this would challenge the misconception that struggling to balance work and family is

a result of personal shortcomings in talent, commitment, or organization, or of being an inadequate parent (McCarver, 2011).

It is evident that transparency on family support at conferences is a major issue. This is a concerning result, but it holds the chance of a very simple improvement of the current circumstances. A simple increase in transparency and easy-to-access information can already improve the conference experience for parents (cf. Appendix A).

The presented survey aimed to gather diverse opinions on the family-friendliness of academic conferences, with the goal

of fostering dialogue between caregivers and event organizers and creating a guide for future improvements. It is important to note, however, that the survey was not designed to be statistically representative of the global scientific community. The target group comprised scientists worldwide, but the distribution pathways – primarily through European conferences and German institutions – led to a focus on European researchers, particularly those from Germany. This is reflected in the fact that more than 50% participants completed the German-language survey. Due to the absence of detailed demographic data on the

global scientific community, we were unable to apply representative weighting to the responses. Childcare systems and family dynamics differ widely across countries, introducing variability that this study could not fully address. Despite these limitations, our work provides a qualitative overview of the current situation, highlighting perspectives from caregivers and sparking a constructive dialogue with conference organizers. Although this study primarily focused on the family responsibilities of parents, we acknowledge that the needs of those caring for elderly family members are equally significant. We hope that this

work will serve as a basis for more inclusive practices in conference planning.

## 5 Conclusions

The presented survey with 245 participants from the geoscientific field provides a balanced view into the difficulties of families in the context of conferences. The participants' responses confirm the barriers for parents in academia already identified in the literature (Jean et al., 2015; Calisi and C. Lopes, 2018; Pohl et al., 2022; Angela L. Bos and Schneider, 2019) and highlight

the difficulties faced by mothers in particular. The answers reveal a broad range of support options and needs. At the same





time, it is evident that it is not only parents who wish for change. Participants without children also welcome a family-friendly transformation and are willing to participate in it.

This work lays the foundation for a dialogue between parents and conference organizers and gives parents a voice in a work environment where family responsibility is still rarely discussed. A broad range of support options can be deduced from these

results that give conference organizers a chance to start somewhere. Even though missing family support is only one of many reasons for the underrepresentation of women in STEM, if implemented well it can significantly help diversify the scientific community. The results of this survey provide conference organizers with clear tasks by calling for more transparency and attention for families in academia. Future work should focus on specifying the guidelines outlined here and on extending them to other subject areas and event formats.

*Data availability.* Anonymized data will be made available on request.

**Appendix A: Guide for conference organizers**

**Before the conference:**

☐ Promote existing offers: Through the website and/or emails, make attendees aware of available services (in detail) so that families/parents feel welcome. This should be communicated well in advance of the conference to give families time to

plan and prepare. Even if there are few or no services available, this should be communicated to ensure transparency and provide parents with a sense of security.

☐ Provide a contact address or phone number: This should be available for questions regarding support for parents/families. It can also be emphasized that conference organizers are willing to find individual solutions.

☐ List of activities: Provide a list of activities that can be done with children in the area (e.g., where the nearest playground

is from the conference center, good places for walks, etc.).

☐ Conference center map: Offer a map of the conference center indicating which areas are accessible with strollers, where customer service, changing tables, nursing rooms, or other (quiet/family) rooms are located.

☐ Clarify registration for children and accompanying persons: Specify in advance if bringing a child to the conference and/or an accompanying person requires registration and if there are any associated costs.

☐ Program markings: Clearly indicate in the program where children are welcome and where they are not.

☐ Forum for parents: Provide a forum where parents can connect and exchange information before the conference.

☐ Selection of family-friendly accommodations: Provide a selection of accommodations that are family-friendly (e.g., offer cribs, so that it is not necessary to book an extra hotel room for the child, which reduces costs).





☐ Survey during registration: Ask during registration if the participant plans to bring children and/or an accompanying person.

☐ Consulting speakers: Ask speakers if they are comfortable with children being in the presentation room. Those who agree should be given guidelines on how to handle situations if children become disruptive.

☐ Family-friendly networking events: Plan networking events that do not take place in the evening and/or where children can be brought along (e.g., with no alcohol served).

☐ Science sessions for children: Scientists could sign up to talk to children of a certain age group about their research for a few minutes in the afternoon.

☐ Financial support for families: Provide information on financial support available for families, including:

- Travel expenses for the child (and accompanying person)

- Additional accommodation costs incurred due to the child (and accompanying person)

- Childcare

- "Participation" in the conference for the child (and accompanying person)

**During the conference:**

☐ Access for parents with strollers: Ensure that parents with strollers have access to all areas.

☐ Family events: Offer events for families (e.g., excursion to a playground).

☐ Children's tables in presentation rooms: Provide tables in presentation rooms for older children, where they can do homework, draw, or solve puzzles during the presentations.

☐ Provide diapers and baby food: These can be offered, possibly for a small fee.

☐ Accessible changing tables: Ensure that changing tables are accessible to everyone.

☐ Nursing rooms: Provide rooms specifically for nursing.

☐ Family rooms: Offer rooms designated for families.

- Detail what facilities are available in these rooms.

- Ensure these rooms are open at all times.

- Provide workspaces for parents or even the possibility to stream presentations from these rooms.

- Offer toys for different age groups.



☐ Education Series on balancing family and academic work: Organize a series of events on the topic of balancing family and academic work, and highlight networks/websites/support services available.

☐ Free access for accompanying persons: Accompanying persons should be able to enter the conference venue free of charge to take care of the child/children.

☐ Childcare services: Provide details about childcare services, including:

– Specify the times during which childcare is available.

     – Indicate the age groups that can be cared for.

     – Specify the language(s) in which childcare is provided.

     – Clarify any costs involved for parents.

☐ Offer the possibility of online conference participation, and consider expanding online networking opportunities.

**After the conference:**

☐ Feedback on family-friendliness: Give parents the opportunity to provide feedback on the family-friendliness of the conference, for example, through a survey that is promoted via email.





**Appendix B:  Survey questions**

We present the questions of the online survey, the response options, question format and target group as well as the number of answers $n$ received for each question in two tables: Tab. B1 lists the questions for demographic information of the survey participants and Tab. B2 lists all other questions.

**Table B1.** Questions regarding the demographic characteristics of the survey participants

| Question | Response Options | Question Format | Target Group | n |
|---|---|---|---|---|
| How old are you? | <15, 15-24, 25-34, 35-44, 45-54, 55-64, 65-74, >74 | Single Choice | All participants | 236 |
| What gender do you identify with? | Male, Female, Non-binary, Other | Single Choice | All participants | 236 |
| By which of these status groups do you feel best described? | Student, PhD, Post-Doc, Lecturer, Staff member, Employed at a university, None | Multiple Choice | All participants | 241 |



**Table B2.** Survey questions

| Question | Response Options | Question Format | Target Group | n |
|---|---|---|---|---|
| At which of these conferences have you already participated in? | EGU, AGU, Ocean Sciences, Other, None | Multiple Choice | All participants | 243 |
| Which other conferences have you attended so far? | - | Free Text Field | All participants | 156 |
| How many conferences and workshops do you typically attend in a year? | 0,1,2-3,3-5,>5 | Multiple choice | All participants | 240 |
| Do you have children? | Yes, No | Filter Question | All participants | 245 |
| How old is your child? | <3,3-6,7-10,11-15,16-18,>18 | Multiple choice | Parents | 141 |
| How important is a hybrid offering to you at conferences and similar events? | very important, important, relatively unimportant, unimportant | Single choice | Parents | 141 |
| Who takes care of your children when you are at a conference? | Partner, Parents, Relatives, Friends, Babysitter, Other, No one | Multiple choice | Parents | 139 |
| Have you ever been unable to attend a conference because of your children? | Yes, No | Single Choice | Parents | 140 |
| Have you ever taken your kids to a conference? | Yes, No , I would like to, I would not like to | Multiple choice | Parents | 134 |
| Which of these offerings might help you at conferences and similar events? | Changing Tables, Rooms to Breastfeed, Playgrounds, Activities, Childcare, Networking, Other, None | Multiple Choice | Parents | 139 |
| What would you wish for to increase a family-friendly environment on conferences or workshops? | | Free Text Field | Parents | 66 |
| Do you generally feel well-informed about support offerings for families from the organizers of conferences? | strongly agree, agree, disagree, strongly disagree | Single Choice | Parents | 143 |
| Do you think that having children is a disadvantage for a scientific career? | strongly agree, agree, disagree, strongly disagree | Single Choice | All participants | 222 |
| Do you feel disturbed by children at conferences and similar events? | strongly agree, agree, disagree, strongly disagree | Single Choice | All participants | 226 |
| Do you feel that conferences and similar events are designed to be family-friendly? | strongly agree, agree, disagree, strongly disagree | Single Choice | Non-parents | 103 |
| Do you think the range of family-oriented offerings at conferences should be expanded? | strongly agree, agree, disagree, strongly disagree | Single Choice | Non-parents | 101 |
| Do you have any ideas on how to better integrate children and families into conferences and similar events? | | Free Text Field | Non-parents | 34 |
| Would you participate in an initiative to showcase your research to children at the conference? | strongly agree, agree, disagree, strongly disagree | Single Choice | Non-parents | 102 |



## Appendix C: Categories from the inductive category formation

The following table (Tab. C1) summarizes the results from the inductive category formation following Mayring and Fenzl (2019). Figure C1 shows the intercode agreement of the categorization performed by two different persons deducing the four main categories "Financial Support", " Awareness", "Clear Communication and Transparency" and "Family Friendly Infrastructure" from the free text answers of the parents in this study to the question "What would you wish for to increase a family-friendly environment on conferences or workshops?". The category "Family Friendly Infrastructure" includes five subcategories (1.1 to 1.5). The percentage in which the categories are addressed in the comments of the parents is shown in Fig. C2.

**Table C1.** Categories from the inductive category formation

| Category Name | Category Number | Absolute Count | % of SUM |
|---|---|---|---|
| **Financial Support** | 2 | 29 | 26 |
| **Awareness** | 3 | 9 | 8 |
| **Clear Communication and Transparency** | 4 | 4 | 3 |
| **Family Friendly Infrastructure** | | | |
| Possibilities for exchange and networking for parents | 1.1 | 2 | 1 |
| Events, where children are explicitly welcomed. | 1.2 | 7 | 6 |
| Family-friendly event schedule | 1.3 | 10 | 9 |
| Childcare/rooms explicitly designed for families. | 1.4 | 26 | 23 |
| Hybrid events | 1.5 | 4 | 3 |

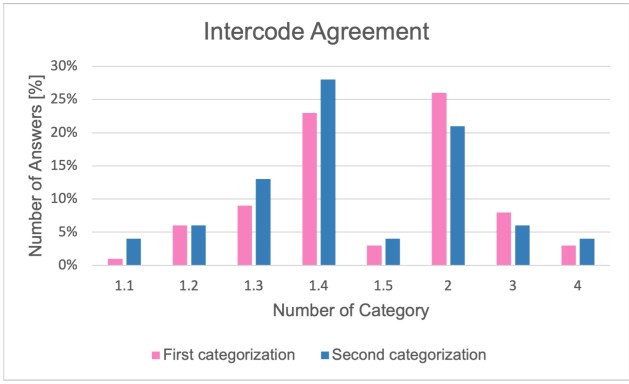

**Figure C1.** Results of the intercode agreement check. Results of the inductive category formation by Elena Päffgen in blue and the results of Sylvia Brückner in orange.



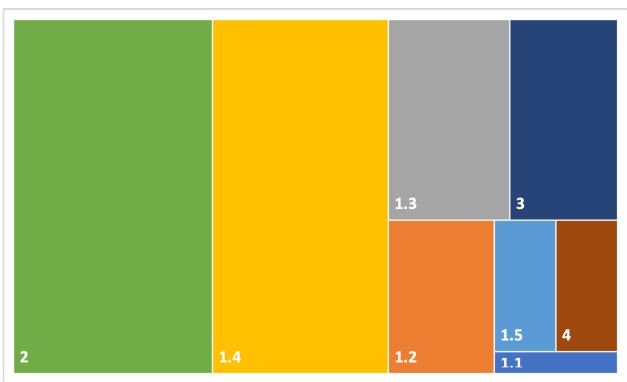

**Figure C2.** Hierarchy plot of the named categories of parents.



*Author contributions.* EP: Writing - Original Draft, Visualization, Formal analysis, Methodology, Investigation; LS & LE: Conceptualization, Writing - Review & Editing, Supervision, Funding acquisition

*Competing interests.* We declare that no competing interests are present.

*Ethical Considerations.* This study was categorized as a low risk study by the Ethics Committee at the Faculty of Medicine Bonn who concluded that there are no ethical or legal concerns to be raised.

*Acknowledgements.* The project was funded by the Office for Equal Opportunities of the University of Bonn. This work was supported by the Open Access Publication Fund of the University of Bonn.We are very grateful for the financial support that made this work possible. We are especially grateful to Manuela Schmidt (University of Bonn, Germany) for her insightful feedback and guidance during the analysis of the survey. Also, we thank Sylvia Brückner (University of Bonn, Germany) for her assistance in analyzing the survey responses. We acknowledge the use of ChatGPT, developed by OpenAI, to streamline the proofreading process, including enhancing grammatical accuracy,
refining sentence structure, and improving flow. Finally, we would like to thank the participants of this survey, whose engagement was invaluable to this study.



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
