# Peer review of "Towards family-friendly conferences in the geosciences: results from a first survey"

_EGUsphere, 2025_

## Author Comment (AC2)

**Response to Comments from Referee #1**

Response to the comments of referee #1 concerning the review of the manuscript "Towards family-friendly conferences in the geosciences: results from a first survey" by Elena Päffgen, Lisa Schielicke, and Leonie Esters submitted to Geoscience Communication (Preprint egusphere-2025-1200).

We thank the referee for their careful reading of our manuscript and for the constructive feedback provided. In the following, the referee's comments are highlighted in **blue** and boldface and our replies will be shown in black below each comment. We have structured our responses according to the organization of the referee's review.

The comments concerning minor language improvements and technicalities will be directly addressed in the manuscript and are not included here.

**General comments:**

> **The title suggests a strong link to geosciences, but this is not evident in the manuscript.**

The title will be changed to "Towards family-friendly conferences: results from a first survey in the geosciences" and the phrase "In the geoscientific field" in line 1 will be removed. This change underlines that the need for family-friendly conferences is not unique to the geoscientific field. Additionally, it encompasses the target group of our study. It also clarifies that generalization of our results to other scientific fields might not be entirely straightforward (this will be shortly discussed in the Discussion section).

> **The manuscript should be improved by making statements more precise, clarifying the scope of cited prior publications, and more thoroughly and quantitatively presenting the analysed survey statistics. At the same time repetitions of key statements should be avoided by reordering the text in some places, e.g. the introduction, and many vague statements need to be expanded and substantiated.**

Concerning this feedback, the following alterations will be made:
- More details concerning the scope and findings of the cited publications will be included in the Introduction (l 37-50).
- Fig. 2 and Fig. 3 will be altered to provide more details on the dataset (shown in New Fig. 2, New Fig.3 below).
  - Fig. 2: The figure will be expanded to show the answers of mothers/fathers/women without children and men without children as well as the age of the children. It will be added that the question did not specify whether online events are included. The subtitle will be changed accordingly.

- ○ Fig.3: Participants without children will be included in the bar plot. The subtitle will be changed accordingly. The shown results will be elaborated on in the text.
- Repetitions of key statements will be removed from the text (e.g. in the Guideline and Discussion section).
- Statements in the introduction, the results and discussion will be expanded and substantiated (see specific comments for details).

[Figure]

New Fig. 2: Answers to the question "How many conferences and workshops do you typically attend in a year?". Only one answer was possible. The type of conference (i.e. online/in-person) was not specified. This question was posed to all participants (245) and skipped by five participants. 240 participants answered this question in total. Panel a shows the responses according to the specified gender and parental status. Panel b shows the answers among the parents (139 in total) further resolved by the age of their youngest child.

[Figure]

New Fig. 3: Answers to the question "Do you think that having children is a disadvantage for a scientific career?". 215 participants answered this question (skipped by 35). The answers are sorted into the indicated gender and whether the participants are parents or not. The shown percentages correspond to the respective group.

**Specific comments**

**Introduction**

> **l 37-50: The statistics presented here could include numbers from the National Report on Early Career Researchers 2025 for Germany, and include more precise statistics for geosciences from the mentioned source for the U.S. (NCSES, table 8.7), including numbers on parents with children under 18, where available. Specifically, a distinction between scientists at universities and research facilities and employees in the private sector could be included to support the argument of conflict between academic career and family planning.**

We thank the referee for pointing us towards existing data on early career researchers. In l. 37-50 the statistics for the U.S. will be included and the mentioned numbers will be presented in more detail (country, scientific background, age of children). Due to the lack of data on the age of first-time fathers in the mentioned countries (e.g., Europe and U.S.), these numbers cannot be included.

The numbers from Metz-Göckel (2014) will be replaced with the 2025 BuWiK statistics on early career researchers in Germany.

In our survey, we did not specifically ask the participants if they worked in the private sector, a research facility or at the university. The private sector is incredibly diverse and encompasses a range of different walks of life that would make it impossible to distill an explanation for any differences found. For these two reasons, we decided against the inclusion of a comparison between women in the academic and private sector. We will clarify this in the text.

> **Consider to include some kind of review of geospecific conferences, larger and smaller events, e.g. EGU, Goldschmidt… where websites are still accessible, with regard to currently implemented family-friendly options., incl. hybrid participation.**

An assessment of the EGU Conference will be included in the appendix based on the information provided by the official website and the conference organizers. Additionally, we will add a figure that provides an overview of the family support in place (see Additional Figure).

We will base our assessment on the information collected during our project work, which is available on our website

https://www.ifgeo.uni-bonn.de/en/sections/meteorology/wg-climate-monitoring/family-friendly-conferences-in-the-geosciences/ranking.

[Figure]

**Family-Friendly Infrastructure at the Conference Venue**

Online participation: Available
Changing tables: Available
Rooms for families: Available
Childcare facilities: Available (Without costs for parents)
Playgrounds near the conference center: Available:
Playground of continents / Sparefroh-Spielplatz in the nearby Donaupark

**Online Resources**

Accessibility information

https://www.egu24.eu/about/accessibility_and_inclusiveness.html

Blogs

https://blogs.egu.eu/geolog/2020/02/19/accessibility-at-egu-parenting-at-the-general-assembly-yes-to-the-creche/

https://blogs.egu.eu/geolog/2022/05/06/how-to-egu22-tips-for-attending-the-conference-with-kids/

https://blogs.egu.eu/geolog/2024/04/12/the-new-egu-colouring-books/

**Financial Support**

Availability of financial support for families/parents: Not directly available
Children under the age of 18 are free of charge but must be accompanied by a legal parent or guardian with a paid registration.
Carers of children under the age of 3 are free of charge, in cases where attendance is only to assist the participation of geoscientist parents.

**Family-Friendly Program**

Sessions/events/activities specifically designed for parents, children, or families: Not available

Sessions/presentations addressing the academia-family conflict or similar topics: Available

Networking events for parents: Available as pop-up events (varies year to year), also EDI reception intended to be family-friendly

**Additional Information**

The EGU Early Career Scientists have a "Life, Careers & Wellness" group which handles topics like work-life balance, including parenting.

**Contact Information**

Point of contact for parents with questions or difficulties: egu24@copernicus.org, egu25@copernicus.org (number changes with the year)

Additional Figure: Overview of the support options offered at the EGU GA, as presented on the "Family-friendly conferences in the geosciences" project website[1]. The information is based on publicly available resources and details provided by the EGU GA organizers; completeness and accuracy cannot be guaranteed.

**Results**

> **l 144-157: The presented aspects are not presented in order of importance and the flow of argumentation is hard to follow and incomplete.**

The aspects presented in l. 144-157 will be ordered according to the importance indicated by the parents in Fig. 5 starting with childcare services. Furthermore, the results will be presented in more detail and the argumentation flow will be improved.

> **l 176-181: Be more specific in terms of the conference setting in describing the lack of awareness and need thereof and elaborate how an increased awareness of the challenge would play out.**

The section currently titled "Awareness" (l. 176-181) will be renamed to "Awareness of the challenges faced by parents in academia" to make the title more specific. Awareness for parents at scientific conferences means recognition of their specific needs by organizers and fellow participants. In the revised section, the lack of awareness will be addressed in the context of conference settings by elaborating on how this issue manifests based on the answers given by the parents and concrete examples. Additionally, we will present how
* * *
[1]

https://www.ifgeo.uni-bonn.de/en/sections/meteorology/wg-climate-monitoring/family-friendly-conferences-in-the-geosciences/ranking

increased awareness among conference organizers and attendees could lead to improvements, such as intentional scheduling, a more inclusive culture that normalizes parental responsibilities within academic networking and participation. Since family life often feels like a deeply personal topic, the challenges of parents often remain hidden (Windsor and Crawford, 2020).

Additionally, we will refer to Feeney and Stritch (2019), who argue that fostering a broader culture of family support is a critical part in enhancing work-life balance in academia. They caution that the mere presence of family-friendly policies might lead to the underuse of these measures.

> **l 229-232: Repetition of introduction aspects and some more references, should be placed there.**

Aspects already presented in the introduction will be removed. Additional arguments presented related to conference guidelines will be supplied with references. For example, the need for flexible solutions is supported by Feeney and Stritch (2019), who found that for women in the US working in the public sector it is important to promote policies that can be tailored to the particular family situation.

**Discussion**

> **Generally, the quantitative results of the survey should be discussed more in detail; are the results surprising etc.,indicative of a specific hypothesis, confirming other observations incl. references.**

We will evaluate our results based on their agreement with each other and connect them to other studies.

For instance, the majority of respondents perceiving children as a disadvantage for an academic career is not surprising, given similar conclusions in the literature (e.g., Metcalfe et al. 2008, Staniscuaski et al. 2023). Our data therefore reinforce the hypothesis that parenthood, particularly motherhood, remains a structural barrier to equal participation in academia (motherhood penalty) (e.g., Goulden et al. 2011, Correll et al. 2007). At the same time, it is notable in our study that men also view children as a disadvantage for an academic career, which aligns with the findings of Cech and Blair-Loy (2019), who analyzed the career trajectories of new parents in STEM fields in the US and found that while mothers are more likely than fathers to leave their profession, parenthood affects the academic career of both genders and is not only a "mother's problem"(Cech and Blair-Loy, 2019, p. 5). Scientists without children expressed strong support for family-friendly measures. This result is somewhat unexpected, as earlier studies often emphasized a lack of awareness among non-parents regarding these challenges (e.g., Ward et al. 2004). This could indicate a growing recognition within the broader academic community of the structural difficulties faced by caregivers.

We will analyze the conference attendance of the participants (Fig. 2) based on the participants parental status and the age of children.

Regarding support measures, we will include the results of Langin (2018) who evaluated the childcare options of 34 scientific conferences. In our study, providing childcare was rated

helpful by 71% of the parents. This aligns with the recommendations of Sardelis et al. (2017) who developed strategies to reduce gender inequities at conferences.

In discussing financial support for families, we will connect our findings to the recommendations by Carter et al. (2024) concerning the design of grants for families.

> **l. 250: Elaborate as to how the survey results show the urgency, e.g. by highlighting aspects using the statistics you presented. Also, your survey is tailored to family support for conferences, which is only part of the academic life.**

We will elaborate on the statement given in l. 250 based on our results concerning the conference participation of the participants and the interference of family and work life. For example, we will include that the majority of interviewed parents could not attend conferences due to family obligations in the past.

Furthermore, while family support at conferences is an important step toward addressing these challenges, the answers of parents also point to the need for broader, systemic measures. In particular, support mechanisms should extend to everyday academic work, during field trips or routine research activities in order to ease the family- work tension in academia.

As one respondent put it: "*The problem is everyday life – too many things don't work out and it costs a lot of organizational time and energy.*"

> **Depending on the disciplines your survey participants come from it would be indicated to narrow "academia" to the field of geosciences to actually link to your title.**
>
> **Please avoid repetitions from the introduction. Instead refer to your results to discuss them in context.**

We will implement a short discussion of the representativeness of our results based on the academic background of the interviewed scientists. The repetitions from the introduction in line 259-262 will be removed and we will ensure that all results are discussed.

**Appendix**

> **Suggestion: A specific example for a conference dedicated to providing family-friendly conditions (e.g. EGU) could be addressed and discussed in terms of the derived guidelines.**

An assessment of the EGU Conference will be added to the Appendix based on the information provided through the website.

> **Suggestion, with respect to "Survey during registration": Before the conference and before registration, e.g. together with abstract submission, requirements for family support could be queried to then tailor the measures to the needs of parenting attendees and communicate these before parents decide to register for attending in person.**

We thank the referee for this valuable suggestion. In l.344, the bulletpoint "Survey during registration…" will be changed to "Survey prior to registration (e.g., during abstract submission): Ask participants at the abstract submission stage whether they require support for bringing children and an accompanying person. This allows the offers to be tailored to parents' needs. Communicate the available measures (including childcare spots and the allocation procedure) before registration so that parents can make an informed decision about attending in person."

> **During the conference: scheduling of sessions in accordance with childcare should be mentioned.**

The following bullet point will be added to the guidelines: "Ensure that the childcare services are in accordance with the session schedule".

> **Transparent communication about the available childcare spots and how they are allocated is missing.**

The transparent communication about the available childcare spots and how they are allocated will be added to the bullet point "Survey prior to registration" (l. 344)

**References**

Statistics (NCSES), N. C. for S. and E. (2022). Doctorate Recipients from U.S. Universities: 2021 (No. NSF 23-300). Article NSF 23-300. https://ncses.nsf.gov/pubs/nsf23300/table/8-7

Metz-Göckel, S., Heusgen, K., Möller, C., Schürmann, R., & Selent, P. (2014). Karrierefaktor Kind: Zur generativen Diskriminierung im Hochschulsystem (1st ed.). Verlag Barbara Budrich. https://doi.org/10.2307/j.ctvddzvkx

BuWiK 2025 National Report on Early Career Researchers 2025 https://buwik.de/en

Windsor, L. C., & Crawford, K. F. (2020). Best Practices for Normalizing Parents in the Academy: Higher- and Lower-Order Processes and Women and Parents' Success. PS: Political Science & Politics, 53(2), 275–280. https://doi.org/10.1017/S1049096519001938

Feeney, M. K., & Stritch, J. M. (2019). Family-Friendly Policies, Gender, and Work–Life Balance in the Public Sector. Review of Public Personnel Administration, 39(3), 422–448. https://doi.org/10.1177/0734371X17733789

Beverly D. Metcalfe, Carol Woodhams, Gina Gaio Santos, Carlos Cabral‑Cardoso; Work‑family culture in academia: a gendered view of work‑family conflict and coping strategies. Gender in Management: An International Journal 22 August 2008; 23 (6): 442–457. https://doi.org/10.1108/17542410810897553

Staniscuaski, F., Machado, A. V., Soletti, R. C., Reichert, F., Zandonà, E., Mello-Carpes, P. B., Infanger, C., Ludwig, Z. M. C., & de Oliveira, L. (2023). Bias against parents in science

hits women harder. Humanities and Social Sciences Communications, 10(1), 201. https://doi.org/10.1057/s41599-023-01722-x

Goulden, M., Mason, M. A., & Frasch, K. (2011). Keeping Women in the Science Pipeline. The ANNALS of the American Academy of Political and Social Science, 638(1), 141–162. https://doi.org/10.1177/0002716211416925

Correll, S. J., Benard, S., & Paik, I. (2007). Getting a Job: Is There a Motherhood Penalty? American Journal of Sociology, 112(5), 1297–1338. https://doi.org/10.1086/511799

Cech, E. A., & Blair-Loy, M. (2019). The changing career trajectories of new parents in STEM. Proceedings of the National Academy of Sciences, 116(10), 4182–4187. https://doi.org/10.1073/pnas.1810862116

Ward, K., & Wolf-Wendel, L. (2004). Academic Motherhood: Managing Complex Roles in Research Universities. The Review of Higher Education, 27(2), 233–257.

Langin, K. (2018). Are conferences providing enough child care support? We decided to find out. Science. https://doi.org/10.1126/science.caredit.aaw3741

Sardelis, S., Oester, S., & Liboiron, M. (2017). Ten Strategies to Reduce Gender Inequality at Scientific Conferences. Frontiers in Marine Science, 4. https://doi.org/10.3389/fmars.2017.00231

Carter, L., Wolz, L., & Pallett, L. J. (2024). Researcher parents are paying a high price for conference travel — here's how to fix it. Nature, 630, 517-518. https://doi.org/10.1038/d41586-024-01571-x

---

## Author Comment (AC3)

**Response to Comments from Referee #2**

Concerning the review of the manuscript "Towards family-friendly conferences in the geosciences: results from a first survey" by Elena Päffgen, Lisa Schielicke, and Leonie Esters submitted to Geoscience Communication (Preprint egusphere-2025-1200).

We thank the referee for their helpful comments and improving our manuscript.
In the following, the referee's comments are highlighted in **blue** and boldface and our replies will be shown in black below each comment.

> **1.  In the category groupings for your results (pg 6): "hybrid" seems to be an odd thing to include with other things that seem to be explicitly about on-site supports. With the importance this option was given by survey participants (76% of survey participants who are parents requesting it), it may be better to have this in it's own category. Isolating it would emphasize it as a strong result and an important outcome.**

We agree with the referee and will add a new category for hybrid meetings to highlight their impact for parents in academia. We will add a dedicated section in line 160 that presents the aspect of online conferences in our survey results.

> **2. One thing I think is missing from the Discussion is the feasibility of acting on the recommendations presented here. For example, one suggestion was to waive the fee for caregivers that accompany conference participants. Financially, this is a reasonable request. The caregivers aren't participating in conference events, just watching a child for an attendee. The cost to the organizers to let these individuals access the conference building at no charge is extremely low. On the other hand, requesting child care while at the same time requesting no fee for children and travel support for families seems to be a bit of a financial paradox. How are the costs for childcare being covered in this scenario – higher fees for everyone? Some other source? In the middle ground is the family-friendly programming. If they are simple events, it is a feasible suggestion that only requires people who are willing to step up and organize the activities and conference organizers who are willing to allocate space for the event. However, more elaborate events that may require supplies or equipment (toys, activities, transportation) have costs involved and again the question arises as to who covers those costs. Then there is the hybrid meeting issue. As someone who champions this idea, the most common reason given for not supporting hybrid participation is cost. At many conference centers, there are outrageous fees associated with hybrid events and it simply costs too much to offer it at a reasonable fee to participants. There are likely other challenges beyond cost that would be easier to deal with. Even without a detailed discussion of these issues, acknowledging that some of these ideas have significant challenges that would need to be addressed would help further the discussion in a constructive way.**

We thank the reviewer for their thoughtful comment. In response, we will incorporate a dedicated section in the "Composition of a guide for conference or workshop organizers"

section (l. 241) that explicitly examines the feasibility of the proposed family-friendly guidelines, with particular attention to the financial and logistical challenges of their implementation. Certain measures, such as waiving fees for caregivers accompanying participants, providing information of the family-friendly infrastructure nearby the conference center (e.g., playgrounds), or family-friendly programming, represent low-cost, achievable measures, as they impose a minimal financial burden to conference organizers and can be facilitated through volunteer engagement. On-site childcare, financial support of families and hybrid event formats require substantial resources that might exceed conference budgets. In our discussion we want to acknowledge these constraints and highlight the measures that can be taken even with low resources to offer organizers realistic options to move towards family-inclusive events.